# Oligometastatic Prostate Cancer: A Comparison between Multimodality Treatment vs. Androgen Deprivation Therapy Alone

**DOI:** 10.3390/cancers14092313

**Published:** 2022-05-06

**Authors:** Francesco A. Mistretta, Stefano Luzzago, Andrea Conti, Elena Verri, Giulia Marvaso, Claudia Collà Ruvolo, Michele Catellani, Ettore Di Trapani, Gabriele Cozzi, Roberto Bianchi, Matteo Ferro, Giovanni Cordima, Antonio Brescia, Maria Cossu Rocca, Vincenzo Mirone, Barbara A. Jereczek-Fossa, Franco Nolè, Ottavio de Cobelli, Gennaro Musi

**Affiliations:** 1Department of Urology, European Institute of Oncology (IEO) IRCCS, 20141 Milan, Italy; francescoalessandro.mistretta@ieo.it (F.A.M.); andrea.conti@ieo.it (A.C.); claudia.collaruvolo@unina.it (C.C.R.); michele.catellani@ieo.it (M.C.); ettore.ditrapani@ieo.it (E.D.T.); gabriele.cozzi@ieo.it (G.C.); roberto.bianchi@ieo.it (R.B.); matteo.ferro@ieo.it (M.F.); giovanni.cordima@ieo.it (G.C.); antonio.brescia@ieo.it (A.B.); ottavio.decobelli@ieo.it (O.d.C.); gennaro.musi@ieo.it (G.M.); 2Department of Oncology and Hemato-Oncology, University of Milan, 20141 Milan, Italy; giulia.marvaso@ieo.it (G.M.); barbara.jereczek@ieo.it (B.A.J.-F.); 3Department of Oncology, European Institute of Oncology (IEO) IRCCS, 20141 Milan, Italy; elena.verri@ieo.it (E.V.); maria.cossurocca@ieo.it (M.C.R.); franco.nole@ieo.it (F.N.); 4Department of Radiotherapy, European Institute of Oncology (IEO) IRCCS, 20141 Milan, Italy; 5Department of Urology, University of Naples Federico II, 80100 Naples, Italy; vincenzo.mirone@unina.it

**Keywords:** prostate cancer, metastatic, prostatectomy, androgen deprivation therapy, radiotherapy

## Abstract

**Simple Summary:**

Data regarding the survival effect of radical prostatectomy in patients with oligometastatic PC (OPC) are sparse and based on small series. Moreover, few studies compared radical prostatectomy with systemic treatment in an OPC setting. We compared multimodality treatment (MMT, defined as robot-assisted radical prostatectomy (RARP) with androgen deprivation therapy (ADT), with or without adjuvant radiotherapy (RT)) vs. ADT alone in oligometastatic prostate cancer (OPC) patients. MMT was associated with lower CSM, mCRPC and second-line therapy rates. A lower rate of treatment-related adverse events was recorded for the MMT group.

**Abstract:**

Background: We compared multimodality treatment (MMT, defined as robot-assisted radical prostatectomy (RARP) with androgen deprivation therapy (ADT), with or without adjuvant radiotherapy (RT)) vs. ADT alone in oligometastatic prostate cancer (OPC) patients. Methods: From 2010 to 2018, we identified 74 patients affected by cM1a-b OPC (≤5 metastases). Kaplan–Meier (KM) plots depicted cancer-specific mortality (CSM), disease progression, metastatic castration-resistant PC (mCRPC), and time to second-line systemic therapy rates. Multivariable Cox regression models (MCRMs) focused on disease progression and mCRPC. Results: Forty (54.0%) MMT and thirty-four (46.0%) ADT patients were identified. On KM plots, higher CSM (5.9 vs. 37.1%; *p* = 0.02), mCRPC (24.0 vs. 62.5%; *p* < 0.01), and second-line systemic therapy (33.3 vs. 62.5%; *p* < 0.01) rates were recorded in the ADT group. No statistically significant difference was recorded for disease progression. ForMCRMs adjusted for the metastatic site and PSA, a higher mCRPC rate was recorded in the ADT group. No statistically significant difference was recorded for disease progression. Treatment-related adverse events occurred in 5 (12.5%) MMT vs. 15 (44.1%) ADT patients (*p* < 0.01). Conclusions: MMT was associated with lower CSM, mCRPC, and second-line therapy rates. A lower rate of treatment-related adverse events was recorded for the MMT group.

## 1. Introduction

Historically, metastatic prostate cancer (PC) has been treated with systemic therapies, namely androgen deprivation therapy (ADT) with or without chemotherapy [1]. However, cytoreductive treatments for patients with metastatic PC have been proposed. The rationale consists of the eradication of castration-resistant and/or lethal tumor cells localized in the prostate gland, which might be the reservoir for metastasis-generating tumor cells [2]. Local treatment survival benefits in patients with low-burden metastatic disease, namely oligometastatic tumors, have been reported for other primary malignancies, such as colorectal cancer, non-small cell lung cancer, and renal cell carcinoma [3,4,5]. More recently, randomized clinical trials, population-based studies, and systematic reviews with metanalysis suggested that local treatment, such as radiotherapy (RT) or radical prostatectomy, have a survival benefit also in patient with metastatic PC [6,7,8,9]. However, data regarding the survival effect of radical prostatectomy in patients with oligometastatic PC (OPC) are sparse and based on small series. Moreover, few studies compared radical prostatectomy with systemic treatment in an OPC setting [10,11,12,13].

Furthermore, retrospective series have suggested that metastasis-directed treatment (MDT) could improve cancer-specific and overall survival with a low toxicity profile [14,15]. In particular, stereotactic body radiotherapy (SBRT) has been described as safe and feasible, allowing the postponement of systemic treatment [16,17,18,19]. Moreover, Steuber et al., reported that combination of SBRT and ADT in OPC patients might improve local control and might decrease the risk of distant failure [20].

Taken together, in the current study, we hypothesized that multimodality treatment (MMT defined as robot-assisted radical prostatectomy (RARP) with extended lymphadenectomy plus ADT, with or without adjuvant local or metastasis-directed RT) might improve survival outcomes in patients with OPC at diagnosis compared to those who received ADT only. Finally, we hypothesized that, compared to ADT alone, patients treated with a multimodality approach might experience fewer treatment-related adverse events during follow-up.

## 2. Materials and Methods

### 2.1. Definition of Population and Variables for Analyses

From July 2010 to July 2018, we retrospectively identified 74 patients affected by cM1a-b OPC (defined as ≤5 metastatic lesions at diagnosis involving non-pelvic lymph nodes (namely M1a) and/or bone (namely M1b)), with locally resectable cT1-T3 tumors. All patients underwent pretreatment staging consisting of a total-body bone scan and a computed tomography scan of the abdomen. In cases of staging uncertainty, choline positron emission tomography was performed to confirm the oligometastatic status. The clinical T-stage assigned after digital rectal examination was successively confirmed by multiparametric magnetic resonance imaging of the prostate in all patients who underwent radical prostatectomy. No patients with visceral metastases at diagnosis were included. The main variable of interest was the type of treatment delivered: MMT vs. ADT alone. MMT patients received RARP with extended pelvic lymphadenectomy and adjuvant ADT (for at least 12 months); patients with pT3, pN1, and/or positive surgical margins were evaluated for adjuvant external beam RT (EBRT), while patients with localized nodal or bone invasion were evaluated for adjuvant stereotactic RT (SBRT). ADT patients did not receive cytoreductive treatment to the primary tumor or metastases.

Descriptive covariates consisted of the following: age; PSA at diagnosis (PSA); Charlson comorbidity index (CCI; ≤2 vs. >2); International Society of Urologic Pathologists grading (ISUP; 1–3, 4–5, or missing data); clinical T-stage (T1-2 vs. T3-4); clinical N-stage (N0 vs. N+); clinical M-stage (M1a vs. M1b); site of metastasis (bone, lymph node, bone plus lymph node). Adjuvant treatment consisted of ADT (intermittent vs. not intermittent) and RT (no RT, adjuvant or salvage EBRT, adjuvant or salvage SBRT, and palliative/symptomatic RT). Second-line systemic therapies were used in patients who developed castration-resistant disease (i.e., Abiraterone acetate, Enzalutamide, chemotherapy). Salvage treatment was defined as EBRT or SBRT performed after 6 months from surgery with the intention to cure a localized progressive disease. Conversely, palliative/symptomatic RT was defined as a treatment delivered to patients with incurable progressive diseases. Oncologic outcomes consisted of metastatic castration-resistant PC (mCRPC, defined as patients with progressive PSA increasing the castration levels of testosterone, with or without radiological disease progression) and disease progression (defined as the radiological diagnosis (obtained or confirmed by either choline positron emission tomography and/or whole-body magnetic resonance) of a novel metastatic lesion with different localizations relative to those present at diagnosis) development, as well as cancer-specific (CSM) and other-cause (OCM) mortality rate. Finally, we reported all adverse events classified as Clavien–Dindo 2 or higher as a rate (occurred vs. not occurred), or we listed them as a specific complication in Appendix A. Patient information is recorded in an approved institutional database (anonymously coded). Moreover, all patients treated at our Institute must sign an informed consent form for records.

### 2.2. Statistical Analyses

First, we analyzed the rates of MMT and ADT alone and tested for possible differences between the two groups. In this step, Chi-squared, Wilcoxon or Mann–Whitney tests estimated differences in proportions or medians. Second, Kaplan–Meier plots depicted differences in CSM, clinical disease progression, development of mCRPC rates, and time to second-line systemic therapy between MMT and ADT alone. Third, MCRMs adjusted for the site of metastasis or PSA focused on disease progression and mCRPC development in MMT vs. ADT alone. All statistical tests were two-sided with a level of significance set at *p* < 0.05. Analyses were performed using the R software environment for statistical computing and graphics (v3.4.1; http://www.r-project.org/ (accessed on 2 May 2020)).

## 3. Results

### 3.1. General Characteristics of the Study Populations and Univariable Treatment Predictors

Out of 74 patients, 40 (54.0%) and 34 (46.0%) underwent MMT and ADT alone, respectively (Table 1). The median PSA level was higher in ADT group relative to the MMT group (87.0 vs. 14.0; *p* < 0.001). No statistically significant differences between the two groups were identified for median age, CCI, ISUP grade, and clinic T-, N-, and M-stages. Although higher rates of bone metastasis (25 vs. 50%) were found in the ADT group, whereas higher rates of lymph node invasion (40 vs. 20.6%) and bone plus lymph node metastasis (35.0 vs. 29.4%) were identified in MMT group, these differences were not statistically significant (*p* = 0.06).

### 3.2. Outcomes after First Therapeutic Line (MMT vs. ADT Alone)

Overall median follow-up was 50 months, 55 vs. 50 for MMT vs. ADT alone (*p* = 0.8), respectively. No statistically significant differences were found for intermittent ADT delivery (47.5 vs. 44.1%; *p* = 1.0). Conversely, higher rates of chemotherapy (CHT) administration were recorded in the ADT group (20.0 vs. 79.4%; *p* < 0.001) relative to MMT.

Radiotherapy was administered differently between MMT and ADT patients (*p* < 0.001). No patient in the ADT group received adjuvant or salvage EBRT, whereas these treatments were delivered in 30.0 and 15.0% of patients in the MMT group, respectively. Metastasis-directed salvage SBRT was administered in 35.0% of MMT patients, while it was not administered in ADT patients. No MMT patients received palliative/symptomatic RT, which was conversely directed to metastatic lesions in 58.8% of ADT patients. No patients in the ADT group underwent a salvage radical prostatectomy. Of all patients, 20 (27.0%) patients experienced a treatment-related adverse event (Appendix A): 5 (12.5%) in MMT and 15 (44.1%) in the ADT group (*p* < 0.01).

### 3.3. Survival Analyses

Of all patients, 22 (29.7%) patients died due to CSM, while 2 (2.7%) patients died due to other causes. Specifically, CSM occurred in 5 (12.5%) and 17 (50.0%) patients of the MMT and ADT-alone group (*p* < 0.001), respectively. Conversely, one patient died due to other causes in each group.

In Kaplan–Meier plots depicting CSM (Figure 1), a higher mortality rate was associated with ADT-alone delivery (5.9 vs. 37.1%; *p* = 0.02).

Similarly, higher mCRPC (24.0 vs. 62.5%; *p* < 0.01; Figure 2) and second-line systemic therapy (33.3 vs. 62.5%; *p* < 0.01; Figure 2) rates were recorded in the ADT group relative to MMT. Conversely, no statistically significant difference was recorded for disease progression, when MMT was compared to the ADT group (83.1 vs. 62.5%; *p* = 0.8; Figure 2).

At MCRMs adjusted for the site of metastasis (Table 2), ADT-alone delivery was associated with a higher rate of mCRPC (Hazard ratio [HR]: 0.40; CI 0.19–0.84; *p* = 0.02). Conversely, no statistically significant difference in treatment delivery effect on disease progression was recorded (HR: 1.19; CI 0.62–2.28; *p* = 0.6).

Similarly, at MCRMs adjusted for PSA (Table 2), ADT-alone delivery was associated with a higher rate of mCRPC (Hazard ratio (HR): 0.39; CI 0.19–0.84; *p* = 0.02). Conversely, no statistically significant difference in treatment delivery effects on disease progression was recorded (HR: 1.05; CI 0.54–2.06; *p* = 0.9). No MCRMs focusing on CSM were performed due to an insufficient number of events.

## 4. Discussion

Although data regarding the survival effect of radical prostatectomy in patients with OPC are limited, recent results are encouraging. We hypothesized that OPC patients who underwent RARP with extended lymphadenectomy plus ADT, with or without local or metastasis-directed RT, might have better oncologic outcomes compared to OPC patients who received ADT only. Moreover, we hypothesized that patients treated with multimodality approach might experience less treatment-related adverse events. Our analyses identified several noteworthy findings.

In the current study, differences in second-line therapy rates have been identified between MMT and ADT groups. Specifically, in Kaplan–Meier plots, a higher second-line treatment (33.3 vs. 62.5%; *p* < 0.01) rate was recorded in the ADT group relative to MMT. Moreover, a higher rate of CHT administration was identified in the ADT group (20.0 vs. 79.4%; *p* < 0.001). The rationale of cytoreductive treatment for delaying systemic therapy has been investigated in previous studies, where cytoreductive or metastasis-directed RT was performed [21]. In the STOMP trial, a longer median ADT-free survival was recorded in patients who received MDT [22]. In this trial, even though oligorecurrent patients were treated (and not upfront OPC), the rationale on the survival benefits of metastasis-directed treatment is similar. Deek et al., considering the better survival outcomes achieved in OPC patients after metastasis-directed RT, hypothesized that MDT might have important clinical significance allowing a delay before the switch to systemic therapy. This result is particularly important in a group of patients whose systemic options can be limited [23]. Taken together, these promising results encourage the delay of second or further-line treatments, lowering the risk for potential adverse effects of systemic therapies and, hopefully, prolonging the time until castration-resistant disease development. In the current study, we recorded a lower rate of castration resistance in patients treated with MMT relative to those treated with ADT only (24.0 vs. 62.5%; *p* < 0.01). Moreover, at MCRMs, ADT delivery was associated with a higher rate of mCRPC (HR: 0.40; *p* = 0.02). Our findings are concordant with those previously published, where longer periods until castration resistance [10] or lower rates of mCRPC [24] were recorded in patients who underwent radical prostatectomy relative to the standard of care. Moreover, of all 74 patients, 20 (27.0%) experienced a treatment-related adverse event, and we identified a lower rate of adverse events in patients treated with MMT (12.5%) relative to the ADT group (44.1%). Specifically, in the ADT group, we recorded cardiovascular events and urinary complications that resulted in palliative surgical procedures (such as ureteral stenting, nephrostomy positioning, and trans-urethral resection of prostate), which did not occur in the MMT group. Moreover, 20 (58.8%) patients of the ADT group underwent palliative or symptomatic RT, while none did in the MMT group. It is noteworthy that the ADT group showed a higher rate of CCI > 3 (52.9 vs. 35%) at baseline relative to the MMT group. Although this difference was not statistically significant, the higher rate of complications and need for palliative or symptomatic RT could be partially attributed to a worse baseline clinical condition. However, the lower rate of adverse events or need for palliative treatment identified in our study is consistent with results previously reported, where a benefit of cytoreductive treatment was recorded when compared to systemic treatment [10,12,13]. In consequence, our results confirm the hypothesis that MMT at diagnosis, compared to ADT alone, prolongs the time to castration-resistant disease development and lowers the risk for potential adverse effects of systemic therapies.

We also investigated differences in cancer-specific survival rates. Specifically, higher actuarial CSM rates were recorded in the ADT group relative to MMT (50.0% vs. 12.5%; *p* < 0.001). In Kaplan–Meier plots, a higher CSM rate was associated with ADT delivery (5.9 vs. 37.1%; *p* = 0.02). Our findings validate previous studies where survival benefits in OPC patients were reported after cytoreductive local treatment, such as radical prostatectomy [6,10,11,13] or RT [9,25], relative to ADT alone. Moreover, our results are consistent with previous studies that reported a survival benefit of MDT in non-localized PC patients [16,20]. This result is emphasized by the lower CSM rate recorded in the MMT group, despite there being no statistically significant differences in disease progression rate. This finding might be explained by the effect on survival of MDT (i.e., SBRT), which was adopted during the follow-up in patients in the MMT group. Taken together, our findings suggest that despite similar disease progression rates between the two groups, the use of MDT after cytoreductive RARP in the case of localized disease progression has improved the overall CSM rate in MMT group. Unfortunately, to the best of our knowledge, there is no study that compares the aforementioned survival outcomes between MMT and the standard of care. In consequence, we could not retrieve any direct comparison with other studies for these specific outcomes.

Overall, our results do not imply the necessity to deliver radical prostatectomy to all OPC patients and do not imply the superiority of radical prostatectomy on other treatment strategies, such as radiotherapy, for the treatment of this specific setting of patients. Instead, the current study supports our initial hypothesis that patients who underwent MMT had better oncologic outcomes relative to those who received ADT alone.

Despite several noteworthy findings, the current study is not devoid from limitations. First, our data represent a retrospective analysis with high potential for selection biases. To maximally reduce biases in survival analyses, we relied on multivariable adjustment. However, it is highly possible that a residual difference persisted according to variables that are unavailable in our database or that could not be tested in multivariable analyses due to the small population. To corroborate univariable analyses, we tested our hypothesis using multivariable models. These models were adjusted for the metastatic site and PSA, which were the two variables that presented the largest difference between groups before treatment. However, we acknowledge that, due to the small population, these multivariable analyses are not entirely reliable. In consequence, we did not base our conclusions on those multivariable models but instead used them only as a validation of univariable Kaplan–Meier results. Nevertheless, it needs to be emphasized that most of studies dealing with cytoreductive radical prostatectomy are based on retrospective analyses conducted on small series. In ours, as well as in all observational retrospective studies, selection biases may threaten the validity of the analysis, and even controlling for covariates by multivariable models or adjusting with propensity score-matching may have a limited effect. Thus, results obtained from these studies should be viewed with caution [26]. In consequence, we need to wait for ongoing randomized clinical trials for definitive conclusions. Similarly, in the near future, several ongoing trials might change the perspective of systemic life-prolonging therapy, expanding the current armamentarium of drugs available for metastatic hormone-sensitive and CRPC patients [27]. Furthermore, no conclusive agreement on OPC definition exists regarding selection biases. We selected M1a-b OPC patients with five or less metastatic lesions according to previous studies, where patients with five or fewer metastatic lesions were associated with better survival rates relative to those with a higher number of metastases [28,29]. The recent HORRAD multi-institutional prospective randomized controlled trial compared irradiation of primary PC with external EBRT with ADT vs. ADT only. This revealed an RT benefit on overall survival only in a subgroup of patients with less than five bone lesions [8,9,25]. Lastly, regarding the adverse event rate, although accessible outside records were reviewed, it is possible that some complications were missed from patients who were lost at follow-up.

## 5. Conclusions

Multimodality treatment was associated with lower CHT and second or further-line systemic therapy rates relative to ADT alone. Moreover, MMT was associated with a lower rate of mCRPC and adverse events. In consequence, it is possible that MMT at diagnosis, compared to ADT alone, prolonged the time until castration-resistant disease development and lowered the risk for potential adverse effects of systemic therapies.

Multimodality treatments were also associated with lower CSM relative to ADT alone, but no statistically significant difference was identified in disease progression rates. Taken together, these findings suggest that, despite similar disease progression rates between the two groups, the use of MMT, particularly the combination of SBRT after cytoreductive RARP with ADT in case of localized disease progression, has improved the overall CSM relative to ADT alone.

## Figures and Tables

**Figure 1 cancers-14-02313-f001:**
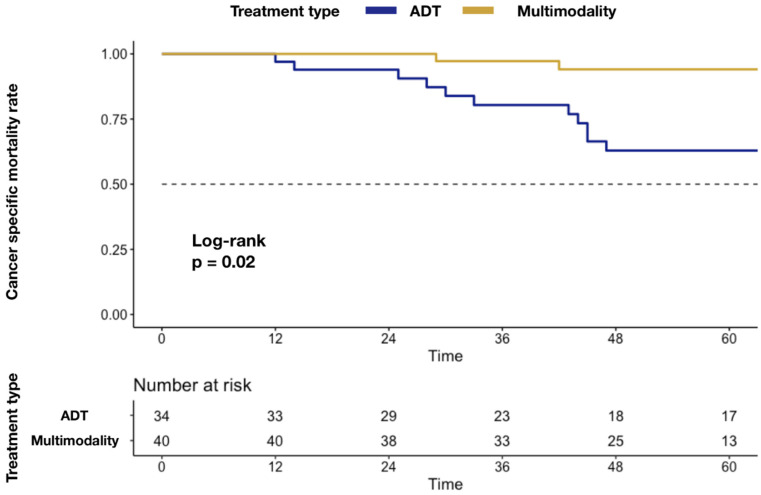
Kaplan–Meier plots depicting differences in CSM rate after stratification according to MMT vs. ADT alone. Abbreviations: cancer specific mortality (CSM); androgen deprivation therapy (ADT); multimodality treatment (MMT); metastatic castration-resistant prostate cancer (mCRPC).

**Figure 2 cancers-14-02313-f002:**
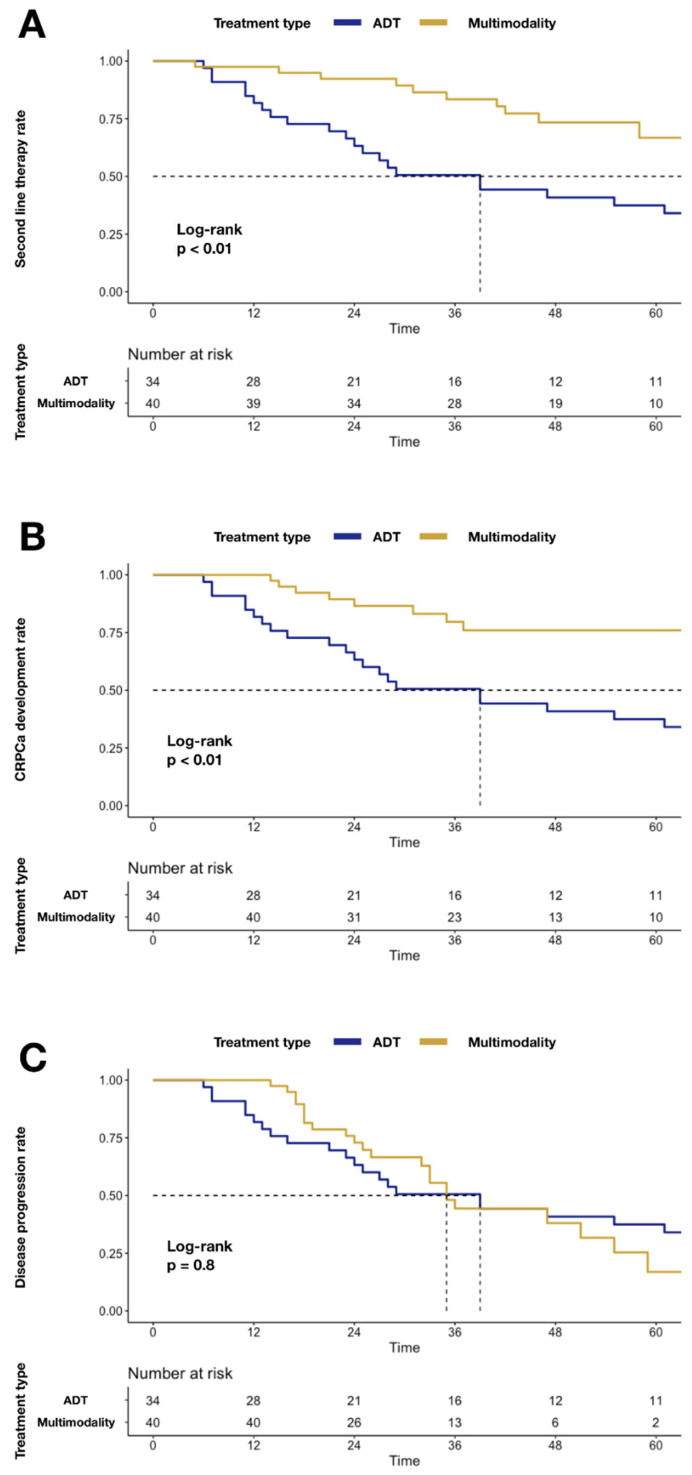
Kaplan–Meier plots depicting differences after stratification according to MMT vs. ADT alone: Panel (**A**) differences in second-line systemic therapy rate; Panel (**B**) differences in mCRPC rate; Panel (**C**) differences in disease progression rate. Abbreviations: cancer-specific mortality (CSM); androgen deprivation therapy (ADT); multimodality treatment (MMT); metastatic castration-resistant prostate cancer (mCRPC).

**Table 1 cancers-14-02313-t001:** Descriptive characteristics of 74 patients with oligometastatic prostate cancer stratified according to androgen deprivation therapy vs. multimodality treatment.

		Overall (74)	ADT Alone (34; 46.0%)	MMT (40; 54.0%)	*p*-Value
Age	Median	66	64	67	0.2
	IQR	58–70	60–74	58–68	
PSA (ng/mL)	Median	23	87	14	<0.001
	IQR	11–85	35–186	9–29	
Charlson comorbidity index	<2	42 (56.8)	16 (47.1)	26 (65.0)	0.2
	≥3	32 (43.2)	18 (52.9)	14 (35.0)	
ISUP grade	1–3	27 (36.5)	9 (26.5)	18 (45.0)	0.1
	4–5	45 (60.8)	23 (67.6)	22 (55.0)	
cT-stage	1–2	32 (43.2)	12 (35.3)	20 (50.0)	0.3
	3–4	42 (56.8)	22 (64.7)	20 (50.0)	
cN-stage	N+	45 (60.8)	20 (58.8)	25 (62.5)	1.0
	N0	29 (39.2)	14 (41.2)	15 (37.5)	
cM-stage	1a	23 (31.1)	7 (20.6)	16 (40.0)	0.1
	1b	51 (68.9)	27 (79.4)	24 (60.0)	
Metastasis site	Lymph node	23 (31.1)	7 (20.6)	16 (40.0)	0.06
	Bone	27 (36.5)	17 (50.0)	10 (25.0)	
	Bone plus lymph node	24 (32.4)	10 (29.4)	14 (35.0)	
Intermittent ADT	No	40 (54.1)	19 (55.9)	21 (52.5)	1.0
	Yes	34 (45.9)	15 (44.1)	19 (47.5)	
Chemotherapy	No	39 (52.7)	7 (20.6)	32 (80.0)	<0.001
	Yes	35 (47.3)	27 (79.4)	8 (20.0)	
Radiotherapy	Salvage SBRT	14 (18.9)	0 (0)	14 (35.0)	<0.001
	Salvage EBRT	6 (8.1)	0 (0)	6 (15.0)	
	Adjuvant EBRT	12 (16.2)	0 (0)	12 (30.0)	
	Palliative/symptomatic RT	20 (27.0)	20 (58.8)	0 (0)	
	No RT	22 (29.7)	14 (41.2)	8 (20.0)	
Adverse events	No	54 (73.0)	19 (55.9)	35 (87.5)	<0.01
	Yes	20 (27.0)	15 (44.1)	5 (12.5)	

**Table 2 cancers-14-02313-t002:** Univariable and multivariable Cox regression models predicting castration-resistant prostate cancer development and disease progression according to treatment delivered (MMT vs. ADT alone). Two separate models are reported. The first model was adjusted for the site of metastasis (defined as bone vs. lymph node vs. bone plus lymph node); the second model was adjusted for PSA (continuously coded).

Models Adjusted for Site of Metastasis							
		Univariable HR	(CI: 2.5–97.5%)	*p* Value	Multivariable HR	(CI: 2.5–97.5%)	*p* Value
Castration-resistant prostate cancer development							
Type of treatment delivered	ADT alone	Ref.			Ref.		
	MMT	0.39	(0.19–0.89)	0.01	0.40	(0.19–0.84)	0.02
Site of metastasis	Lymph node	Ref.			Ref.		
	Bone	1.59	(0.64–3.91)	0.3	1.16	(0.46–2.95)	0.7
	Bone plus lymph node	1.66	(0.67–4.12)	0.3	1.46	(0.58–3.64)	0.4
Disease progression							
Type of treatment delivered	ADT alone	Ref.			Ref.		
	MMT	1.09	(0.60–1.99)	0.8	1.19	(0.62–2.28)	0.6
Site of metastasis	Lymph node	Ref.			Ref.		
	Bone	1.25	(0.60–2.61)	0.5	1.35	(0.62–2.96)	0.5
	Bone plus lymph node	1.46	(0.70–3.03)	0.3	1.51	(0.72–3.18)	0.3
**Models Adjusted for PSA**							
		**Univariable** **HR**	**(CI: 2.5–97.5%)**	***p* Value**	**Multivariable** **HR**	**(CI: 2.5–97.5%)**	***p* Value**
Castration-resistant prostate cancer development							
Type of treatment delivered	ADT alone	Ref.			Ref.		
	MMT	0.39	(0.19–0.81)	0.01	0.39	(0.18–0.85)	0.02
PSA	Continuously coded	1.00	(0.99–1.01)	0.2	1.00	(0.99–1.01)	0.9
Disease progression							
Type of treatment delivered	ADT alone	Ref.			Ref.		
	MMT	1.09	(0.60–1.99)	0.8	1.05	(0.54–2.06)	0.9
PSA	Continuously coded	1.00	(0.99–1.01)	0.7	1.00	(0.99–1.01)	0.8

## Data Availability

The data presented in this study are available on request from the corresponding author.

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
