# Peer review of "Oligometastatic Prostate Cancer: A Comparison between Multimodality Treatment vs. Androgen Deprivation Therapy Alone"

_cancers, 2022, doi:10.3390/cancers14092313_

Round 1

Reviewer 1 Report

The authors are promoting what they term "multimodality treatment" (MMT) and seem to be confident that there is an important role for including a radial prostatectomy (RP).  As we have good data that the RP adds anything to RT & ADT the authors should acknowledge this fact.  The fact is (as supported by the two references attached) there is huge selection bias as to which patients get surgery and which don't and why.  They authors exaggerate the significance of their findings and essentially are implying "everyone should get an RP" 

Author Response

Reviewer 1

The authors are promoting what they term "multimodality treatment" (MMT) and seem to be confident that there is an important role for including a radial prostatectomy (RP). 

Comment 1: 

As we have good data that the RP adds anything to RT & ADT the authors should acknowledge this fact. 

Answer: We thank the reviewer for the pertinent comment. We do agree with the reviewer that, relative to radiation therapy plus hormone therapy, data regarding the treatment of oligometastatic PCA patients are limited, sparse and based on small populations. Thus, we reported this aspect both in the Introduction (“However, data regarding the survival effect of radical prostatectomy in patients with oligometastatic PC (OPC) are sparse and based on small series. Moreover, only few studies compared radical prostatectomy with systemic treatment, in OPC setting.”) and in the Discussion section (“…data regarding the survival effect of radical prostatectomy in patients with OPC are limited…”). However, despite all the limitations of these studies dealing on the topic (i.e. retrospective nature, small populations and/or selection biases), their results are encouraging as demonstrated by Heidenreich et al., Gratzke et al., Stauber et al., whose studies were cited in the original version of the manuscript. However, our hypothesis was not to demonstrate the validity of radical prostatectomy in this setting, but to validate the effect of a multimodality treatment, in which radical prostatectomy is only part of the workout, and has been integrate with radiation therapy and a period of hormone therapy. This particular aspect was clarified in the Discussion section as result of the answer of your Comment 3 (read later).

Comment 2: 

The fact is (as supported by the two references attached) there is huge selection bias as to which patients get surgery and which don't and why. 

Answer: We thank the reviewer for the pertinent comment. We agree with the reviewer’s comment and, in fact, even in the first version of the manuscript we tried to highlight all these biases in the limitation paragraph, shading lights on how we tried to smooth the effect of these biases controlling for covariates with multivarable models. However we agree that a detrimental affect may still exist. In consequence, to comply with all the reviewer’s suggestions, we corrected the Discussion’s section that now reads as follows: “In our, as well as, in all observational retrospective studies selection biases may threaten the validity of the analyses, and even controlling for covariates by multivariable models or adjusting with propensity score matching may have a limited effect. Thus, results obtained from these studies should be viewed with caution (Ref. Limits of Observational Data in Determining Outcomes From Cancer Therapy).”.

Comment 3: 

They authors exaggerate the significance of their findings and essentially are implying "everyone should get an RP" 

Answer: We thank the reviewer for the pertinent comment. We apologize if our message was misleading. We absolutely do not think that all oligometastatic PCa patients should undergo radical prostatectomy. And even less we do not claim any superiority of radical prostatectomy on other treatment strategies, such as radiotherapy. On contrary we just wanted to report our data on a particular series of oligometastatic PCa treated with a multimodality approach. To clarify our message we added the following passage to the Discussion section: “Taken together, our results do not imply the necessity to deliver radical prostatectomy to all OPC patients, and even less do not imply the superiority of radical prostatectomy on other treatment strategies, such as radiotherapy, for the treatment of this specific setting of patients. Instead, the current study supports our initial hypothesis that patients who underwent MMT had better oncologic outcomes relative to those who received ADT alone.”

Reviewer 2 Report

In their retrospective analysis, Mistretta et al investigate the effect of multimodality treatment for cM1a-c oligometastatic prostate cancer on cancer specific mortality, disease progression, development to metastatic castration resistant prostate cancer and time to second-line systemic therapies, compared to ADT alone.

The manuscript is very well written and reads fluently.

Although the retrospective design is a major drawback to make conclusions on this topic, the analysis is at least hypothesis generating and provides useful information for the readers. The authors note very well the limitations of this investigation.

Some suggestions:

  • introduction, page 1 line 11: "might have a survival benefit": in my opinion, considered the data from STAMPEDE (Parker et al) for patients with low metastatic burden, this statement may be expressed 'stronger'
  • introduction, page 2 line 11: please add "with or without adjuvant radiotherapy" to be consistent and complete
  • materials and methods, page 2: as you report on adverse effects, information on the interpretation/scoring is lacking
  • results page 4: please add range/CI , according to the preference of the journal
  • results page 4 section 3.2: if you add salvage (SB)RT and adjuvant EBRT to the table (0 versus X), I would also add radical prostatectomy, to make the overview transparant and complete
  • low toxicity in MMT group, even though quite some patients received ART/SRT: can you provide the reader with more information on which toxicity scoring was used (see methods, cfr supra), was grade 1-2 toxicity included?
  • results: information on the duration of ADT in the MMT group is lacking
  • discussion: comparison with STOMP might be confusing for the reader, please elaborate a bit more on the different disease settings (upfront versus oligorecurrent disease) to correctly interpret the data
  •  lower rate of AE and need for palliative treatments is a very important message to the readers in my opinion. how many patients did need palliative treatment/antalgic radiotherapy for local progression?
  • major limitation is the retrospective design and selection bias, however the authors state this very clear

Author Response

Reviewer 2

In their retrospective analysis, Mistretta et al investigate the effect of multimodality treatment for cM1a-c oligometastatic prostate cancer on cancer specific mortality, disease progression, development to metastatic castration resistant prostate cancer and time to second-line systemic therapies, compared to ADT alone.

The manuscript is very well written and reads fluently.

Although the retrospective design is a major drawback to make conclusions on this topic, the analysis is at least hypothesis generating and provides useful information for the readers. The authors note very well the limitations of this investigation.

Some suggestions:

Comment 1: 

Introduction, page 1 line 11: "might have a survival benefit": in my opinion, considered the data from STAMPEDE (Parker et al) for patients with low metastatic burden, this statement may be expressed 'stronger'

Answer: We thank the reviewer for the pertinent comment. To comply with the reviewer’s suggestion, we modified the sentence that now reads as follows: “More recently, randomized clinical trials, population-based studies and systematic reviews with metanalysis suggested that local treatment, such as radiotherapy (RT) or radical prostatectomy, have a survival benefit also in patient with metastatic PC”.

Comment 2: 

Introduction, page 2 line 11: please add "with or without adjuvant radiotherapy" to be consistent and complete

Answer: We thank the reviewer for the pertinent comment. To comply with the reviewer’s suggestion, we modified the sentence that now reads as follows: “Taken together, in the current study we hypothesised that multimodality treatment (MMT defined as: robot-assisted radical prostatectomy (RARP) with extended lymphadenectomy plus ADT, with or without adjuvant local or metastasis-directed RT) might improve survival outcomes in patients with OPC at diagnosis compared to who received ADT only.”

Comment 3: 

Materials and methods, page 2: as you report on adverse effects, information on the interpretation/scoring is lacking

Answer: We thank the reviewer for the pertinent comment. In the current analyses we reported all adverse events classified as Clavien-Dindo 2 or higher occurred and reported them as rate (occurred vs. not occurred) or listing them as specific complication occurred. To comply with the reviewer’s suggestion we modified the Materials and Methods section as follows: “Last, we reported all adverse events classified as Clavien-Dindo 2 or higher occurred and reported them as rate (occurred vs. not occurred) or listed as specific complication occurred in Supplementary Table 1”

Comment 4: 

Results page 4: please add range/CI , according to the preference of the journal

Answer: We thank the reviewer for the pertinent comment. We corrected the manuscript as requested.

Comment 5: 

Results page 4 section 3.2: if you add salvage (SB)RT and adjuvant EBRT to the table (0 versus X), I would also add radical prostatectomy, to make the overview transparant and complete.

Answer: We thank the reviewer for the pertinent comment. No ADT patients underwent a salvage radical prostatectomy. To comply with the reviewer’s suggestion we modified the Results section 3.2 adding the following passage: “No patients in ADT group underwent a salvage radical prostatectomy”.

Comment 6: 

Low toxicity in MMT group, even though quite some patients received ART/SRT: can you provide the reader with more information on which toxicity scoring was used (see methods, cfr supra), was grade 1-2 toxicity included?

Answer: We thank the reviewer for the pertinent comment. Due to the miscellaneous nature of adverse events that can occur when more then one treatment (of different nature such as surgery, RT or ADT) are delivered we did not used the RT toxicity classification, but reported all the adverse events occurred that could be classified as Clavien Dindo scale grade two or higher. To avoid misleading messages, in the whole manuscript we used the word “toxicity” just once in the Introduction section (in the following passage: “Furthermore, retrospective series have suggested that metastasis-directed treatment (MDT) could improve cancer specific and overall survival, with low toxicity profile.”), and whether needed we replaced words such as “toxic” or “toxicity”. To comply with your Comment 3, we modified the Materials and Methods section as follows: “Last, we reported all adverse events classified as Clavien-Dindo 2 or higher occurred and reported them as rate (occurred vs. not occurred) or listed as specific complication occurred in Supplementary Table 1”. 

Comment 7: 

Results: information on the duration of ADT in the MMT group is lacking.

Answer: We thank the reviewer for the pertinent comment. To comply with the reviewer’s suggestion we modified the Materials and Methods section as follows: “MMT patients received RARP with extended pelvic lymphadenectomy and adjuvant ADT (for at least 12 months)”

Comment 8: 

Discussion: comparison with STOMP might be confusing for the reader, please elaborate a bit more on the different disease settings (upfront versus oligorecurrent disease) to correctly interpret the data.

Answer: We thank the reviewer for the pertinent comment. To comply with the reviewer’s suggestion we modified the Discussion section as follows: “In the STOMP trial a longer median ADT-free survival was recorded in patients who received MDT.22 In this trial, despite that oligorecurrent patients were treated, and not upfront OPC, the rationale on survival benefit of metastasis-directed treatment is similar.”

Comment 9: 

Lower rate of AE and need for palliative treatments is a very important message to the readers in my opinion. how many patients did need palliative treatment/antalgic radiotherapy for local progression?

Answer: We thank the reviewer for the pertinent comment. We do agree that this outcome, in association with the adverse events rate, is very important and complementary to the oncologic results usually reported in this kind of manuscript. As reported in the Results section 3.2 and in the Table 1, 20 (58.8%) patients of the ADT group underwent palliative or symptomatic RT, while none in the MMT group. To stress this aspect and to comply with the reviewer’s suggestion we modified the Discussion section as follows: “Specifically, in ADT group we recorded cardiovascular events and urinary complications that led to palliative surgical procedures (such as ureteral stenting, nephrostomy positioning and trans-urethral resection of prostate), which did not occur in the MMT group. Moreover, 20 (58.8%) patients of the ADT group underwent palliative or symptomatic RT, while none in the MMT group. The lower rate of adverse events or need for palliative treatment identified in our study is consistent with similar results previously reported, where a benefit of cytoreductive treatment was recorded when compared to systemic treatment.10,12,13 In consequence, our results confirm the hypotheses that MMT at diagnosis, compared to ADT alone, prolongs the time to castration resistant disease development and lower the risk for potential adverse effect of systemic therapies.”

Comment 10: 

Major limitation is the retrospective design and selection bias, however the authors state this very clear.

Answer: We thank the reviewer for the comment.

Reviewer 3 Report

oligometastatic prostate cancer: a comparison between multi modality treatment vs standard or care, is a retrospective review of synchronous oligmetastatic prostate cancer treated with iADT or cADT vs RARP + ADT +/- RT). Although there are a number limitation/flaws in the design and analysis of the study, given the nature/scope of the manuscript and recommendation, they dont need to be enlisted here. These are the reasons why the manuscript does not merit publication:

  1. for such patient the standard of care has no evolved, and it includes RT to prostate (STAMPEDE) cADT and ARATs (LATITUDE, ENZAMET, TITAN)
  2. the study does not add to scientific knowledge

Author Response

Reviewer 3

oligometastatic prostate cancer: a comparison between multi modality treatment vs standard or care, is a retrospective review of synchronous oligmetastatic prostate cancer treated with iADT or cADT vs RARP + ADT +/- RT). Although there are a number limitation/flaws in the design and analysis of the study, given the nature/scope of the manuscript and recommendation, they dont need to be enlisted here. These are the reasons why the manuscript does not merit publication:

-for such patient the standard of care has no evolved, and it includes RT to prostate (STAMPEDE) cADT and ARATs (LATITUDE, ENZAMET, TITAN)

-the study does not add to scientific knowledge

Answer: We thank the reviewer for the comment.

Round 2

Reviewer 1 Report

The authors should acknowledge that the higher complication rates in the ADT group could mean they were just sicker in the first place, that is why they had no local treatment.  

Author Response

Comment 1: 

The authors should acknowledge that the higher complication rates in the ADT group could mean they were just sicker in the first place, that is why they had no local treatment. 

Answer: We thank the reviewer for the pertinent comment. We reported data on Charlson comorbidity index, as showed in table 1. After comparison we found a higher rate of CCI > 3 in the ADT group (52.9 vs. 35%), however this difference was not statistically significant. In consequence, it is possible that the higher rate of complications and need for palliative or symptomatic RT could be partially attributed to a worst baseline clinical condition. To comply with the reviewer’s suggestion, we modified the Discussion section as follows: “It is noteworthy that ADT group showed a higher rate of CCI > 3 (52.9 vs. 35%) at baseline, relative to MMT group. Despite this difference was not statistically significant, the higher rate of complications and need for palliative or symptomatic RT could be partially attributed to a worst baseline clinical condition. However, the lower rate of adverse events or need for palliative treatment identified in our study is consistent with similar results previously reported, where a benefit of cytoreductive treatment was recorded when compared to systemic treatment.10,12,13 In consequence, our results confirm the hypotheses that MMT at diagnosis, compared to ADT alone, prolongs the time to castration resistant disease development and lower the risk for potential adverse effect of systemic therapies.”.

Reviewer 3 Report

n/a

Author Response

We thank the Reviewer